# HRT in Women Undergoing Pelvic Clearance for Endometriosis—A Case Report and a National Survey

**DOI:** 10.3390/jcm12010336

**Published:** 2023-01-01

**Authors:** Saad Amer, Subul Bazmi

**Affiliations:** 1Division of Translational Medical Sciences, School of Medicine, Royal Derby Hospital Centre, University of Nottingham, Derby DE22 3DT, UK; 2Department of Obstetrics and Gynaecology, University Hospitals of Derby and Burton NHS Foundation Trust, Derby DE22 3NT, UK

**Keywords:** HRT, menopause, endometriosis, hysterectomy, chronic pelvic pain

## Abstract

The optimal hormone replacement therapy (HRT) in women who have undergone pelvic clearance for endometriosis remains uncertain with insufficient evidence. The purpose of this case report and the national survey was to highlight the potential HRT-related risks and to establish current HRT practice in this group of women. The case was a 45-year-old woman presenting with recurrence of severe chronic pelvic pain while on oestrogen-only HRT (EO-HRT) for five years after subtotal hysterectomy and bilateral oophorectomy for severe endometriosis. MRI revealed multiple peri-cervical endometriomas and severe right hydroureter/hydronephrosis with complete right renal parenchymal loss. The survey was a 21-item questionnaire administered electronically using SurveyMonkey. It was reviewed and approved by British Menopause Society and British Society of Gynaecological endoscopy and was sent to their members as well as NHS Gynaecologists. A total of 216 physicians responded including 120 (55.6%) Gynaecology Consultants and 96 (44.4%) GPs/Nurses in Menopause clinics. Overall, 68.6% of responders prescribe combined HRT (C-HRT), 11.1% tibolone, 13.0% EO-HRT and 7.8% varied HRT. Fifty-one percent prescribe the progestogen component of C-HRT indefinitely, 22% for 3–6 months and 27% for varied durations. In conclusion, this study highlights the real risk of endometriosis recurrence in EO-HRT users after pelvic clearance for endometriosis. The survey revealed that only two thirds of Gynecologists/Menopause practitioners prescribe combined HRT in this group of women.

## 1. Introduction

Endometriosis is a very common gynaecological condition affecting 10% of women of reproductive age and is often associated with severe and debilitating chronic pelvic pain (CPP). It is characterized by the presence of endometrial-like tissue outside the uterine cavity, which induces chronic inflammation and fibrosis. It is usually found in the pelvis, but frequently affects extra pelvic sites such as the diaphragm, lungs, brain, and skin.

Treatment strategies usually aim to alleviate pain and suppress or eradicate the endometriotic lesions. Current management options include pain control, suppressive hormonal therapy, or surgery (conservative or radical). With a high recurrence rate (~50%) after conservative surgery, which involves complete excision of endometriosis without hysterectomy [1,2], up to 37% of premenopausal women will eventually undergo the radical surgery in the form of total hysterectomy and bilateral salpingo-oophorectomy (TH-BSO) as a more definitive and permanent solution [3]. This approach induces iatrogenic premature or early menopause in these women who will need hormone replacement therapy (HRT) for many years until at least their natural age of menopause [4,5].

The main concern with HRT after pelvic clearance for endometriosis is the potential risk of reactivation or malignant transformation of residual endometriosis foci due to exposure to exogenous oestrogen. A large population study has reported 8% reoperation rate for recurrent endometriosis after TH-BSO increasing to 19% if the ovaries were conserved [6]. Another previous study reported a high recurrence rate of 62% after hysterectomy in advanced stages of endometriosis if the ovaries were conserved [7]. The authors concluded that ovarian conservation was associated with 6-fold increase in risk of recurrent pain and 8-folds risk of reoperation.

Recurrence of endometriosis after pelvic clearance has been reported in many pelvic as well as extra-pelvic sites resulting in a wide range of possible symptoms. A systematic review by Gemmell et al. reported endometriosis recurrence in the genitourinary (bladder, ureter, ovary, cervix, and vagina), gastrointestinal (bowel and rectum) and pulmonary systems (lung and bronchi) [8]. Symptoms depend on the sites of recurrence and commonly include pain (pelvic and extra-pelvic), bleeding (e.g., vaginal bleeding, haematuria, rectal bleeding and haemoptysis) and pelvic masses [8].

Gemmell et al. reported that most post hysterectomy recurrences of endometriosis occurred in women receiving HRT especially when EO-HRT was used [8]. Furthermore, the systematic review reported that HRT was also associated with a risk of malignant transformation [8]. The authors therefore recommended that EO-HRT should be avoided in women undergoing pelvic clearance for endometriosis. This has led the British Menopause Society (BMS) and European Society of Human Reproduction and Embryology (ESHRE) to recommend continuous C-HRT and avoidance of EO-HRT in this group of women at least until age of natural menopause [9,10]. BMS specified that the C-HRT should be considered in women following hysterectomy for severe endometriosis to prevent reactivation of residual disease and to potentially prevent malignant transformation of residual deposits. The quality of evidence for ESHRE recommendations to use C-HRT and to avoid EO-HRT were deemed low, while the recommendation to continue C-HRT until natural age of menopause was rated as good practice point [10]. Similarly, the authors of the BMS guidelines acknowledged the limited evidence available on this to guide clinical practice [9]. Therefore, optimal HRT in young women who have undergone pelvic clearance for endometriosis remains uncertain.

The purpose of the case report in this study was to highlight the potential risk associated with EO-HRT after pelvic clearance for severe endometriosis. The purpose the national survey was to establish current UK practice in the light of current guidelines and to raise awareness amongst all Gynecologists and GPs regarding the main issues and risks associated with HRT after pelvic clearance for endometriosis.

## 2. Materials and Methods

### 2.1. Case Report

We present a case of recurrence of severe endometriosis symptoms associated with oestrogen-only HRT in a 45-year-old woman after subtotal hysterectomy and bilateral salpingo-oophorectomy (STH-BSO) for severe endometriosis. Consent was obtained from the patient for using her anonymized clinical information including images in this report. Hospital records were reviewed for clinical, radiological, operative and pathological findings.

### 2.2. National Survey

#### 2.2.1. Study Design and Population

This was a cross-sectional survey of clinicians and practitioners managing surgical menopause in women after pelvic clearance for endometriosis. Pelvic clearance was defined as total hysterectomy and bilateral salpingo-oophorectomy (TH-BSO) with excision of endometriosis.

#### 2.2.2. Questionnaire Design and Distribution

The survey was developed by the authors and then revised following peer-review by representatives of British Menopause society (BMS) and British Society of Gynaecological endoscopy (BSGE). It was approved by both societies and sent to their members as well as to other NHS Gynecologists. Participants were e-mailed an introductory letter and a link to the electronic survey. Two reminders were later sent if there was no response to the initial invitation.

The survey comprised of a 21-item questionnaire administered electronically using SurveyMonkey software, (www.surveymonkey.com), which is an internet-based survey tool for collection and analysis of responses. The survey focused on various controversial issues encountered when prescribing HRT after inducing surgical menopause in women with endometriosis, for which no clear evidence could be drawn from the existing scientific literature. The initial questions were designed to ensure that responders completing the survey have the relevant experience, while screening out others. Further questions mainly focused on HRT in different age groups, type of HRT (C-HRT, EO-HRT or tibolone), timing of HRT commencement after surgery, type and duration of the progestogen component of C-HRT and the rational for the choice of HRT. Other questions included indications for C-HRT and OE-HRT, type of oestrogen and progestogen used in HRT, alternatives to HRT and holistic/lifestyle measures to tackle menopause.

## 3. Results

### 3.1. Case Report

#### 3.1.1. Presentation

A 45-year-old woman was referred to Derby endometriosis centre, UK in February 2021 with a 2-year history of severe and debilitating CPP while on high dose EO-HRT (Estradiol Valerate 3 mg/day) for five years after STH-BSO for severe endometriosis in a different hospital. The reason given by the previous surgeon, who was not an endometriosis specialist, for performing subtotal rather than total hysterectomy was the extensive deep peri-cervical endometriosis. It is therefore clear that this surgery did not involve complete excision of endometriosis leaving behind significant peri-cervical disease. Following surgery. the patient was started on EO-HRT initially 2 mg/day, which was later increased to 3 mg/day to control her severe menopausal symptoms. She was followed by her surgeon. Initially, she was pain-free and remained so for around three years when she started to experience recurrence of pain. The pain gradually worsened until it has become severe and debilitating after about four years from surgery. At that point, she was referred to Derby Endometriosis Centre.

Pelvic examination revealed tender and fixed cervical stump with no palpable endometriosis nodules. An MRI of the pelvis revealed multiple endometriomas over the cervical stump with the largest measuring 3 cm in diameter and with extensive adhesions to the sigmoid (Figure 1). There was also severe right hydroureter and hydronephrosis with complete right renal parenchymal loss. There was no deep disease infiltrating the bowel and the left ureter was normal. Further assessment confirmed complete failure of the right kidney and the option of right nephrectomy was discussed with the patient. The MRI images were reviewed and discussed with the Endometriosis Centre Radiologist who ruled out possibility of malignancy.

#### 3.1.2. Management

Given the severity of symptoms, surgical management was offered to the patient including trachelectomy and excision of the endometriomas. The patient agreed and was added to the waiting list for laparoscopy. In the meantime, her HRT was modified by reducing the estradiol valerate dose to 2 mg/day and adding Norethisterone tablets 15 mg/day. HRT was prescribed to alleviate severe menopausal symptoms and to provide long-term protection for bone health. The new regime moderately improved her pelvic pain. Interestingly, when she was later reviewed in her original hospital, the clinicians stopped the progestogen component explaining that it was not necessary as she did not have a uterus.

Laparoscopy was later performed by the Endometriosis Gynecologist jointly with the Colo-Rectal surgeon. This revealed a completely obliterated pelvis the sigmoid covering and adherent to the cervical stump, pelvic walls and the bladder (Figure 2a). Extensive adhesiolysis was carried out with complete mobilization of the recto-sigmoid (Figure 2b). The endometrioma complex, which was attached to the top of the cervical stump, were dissected from all the surrounding adhesions (Figure 2c). Colpotomy was then performed with complete excision of cervix and the attached endometriomas (Figure 2d). Following surgery, she was put back on continuous C-HRT. Six weeks after surgery, she reported good recovery with complete disappearance of her pelvic pain. Histology of the excised cervix and surrounding lesions showed densely fibrotic tissue covered with florid endometriosis with numerous disordered endometrial glands surrounded by prominent decidualized stroma in keeping with progestogen effect. There was no atypia or malignancy. The cervix showed endometrioid adenomyomas with a prominent decidualized endometrium with no atypical or malignant changes.

### 3.2. National Survey

#### 3.2.1. Responders

A total of 216 (after exclusion of one irrelevant responder) clinicians/practitioners responded to the survey with 70% completion rate. Responders included 120 (55.6%) Gynaecology Consultants, 85 (39.4%) Primary Care Menopause Practitioners (GPs and Reproductive Health Practitioners) and 11 (5.0%) Nurse Practitioners in menopause or reproductive health (Figure 3a). The Gynaecology Consultants included 47 (21.8%) Endometriosis Specialists, 50 (23.2%) General Gynecologists and 23 (10.6) Gynaecologists with interest in menopause.

Approximately, 75% of responders had over 10 years of experience with 98% practicing in the NHS (Figure 3b). Over 55% of responders manage > 10 women annually after pelvic clearance for menopause.

#### 3.2.2. Age of Women and HRT

Overall, 89.6%, 82.6%, 61.8% and 29.3% of responders prescribe HRT to all women aged <40, 40–45 and over 50 years, respectively after pelvic clearance for endometriosis.

There was an increasing percentage of responders not prescribing HRT at all with the increasing age of women from 2.1% in women under 40 to 23.9% in over 50 years. Similarly, the percentages of responders leaving the decision of HRT to the woman ranged from 7.6% in women <40 to 45.6% in women >50 years (Figure 4).

#### 3.2.3. Type of HRT Prescribed

Overall, 68.6% of the responders prescribe combined HRT, 13.0% oestrogen only, 11.1% Tibolone, and 7.8% varied HRT depending on endometriosis severity and completeness of excision (Figure 5a). When divided according to grade, combined HRT or Tibolone is prescribed by 81.6% of Endometriosis specialists, 70.8% of Gynaecology consultants, 45% of Gynae consultants with interest in Menopause and 81.4% of GPs/CNS with interest in Menopause (Figure 5b). Comparison between different grades using Q-square test revealed a statistically significant (<0.05) difference between Gynaecology consultants and GP/CNS/APN in prescribing Tibolone (25.0% versus 2.3%, respectively). No other statistically significant difference was observed between grades. EO-HRT is prescribed by 7.9% of Endometriosis specialists, 12.5% of Gynaecology consultants, 25% of Gynae consultants with interest in Menopause and 4.7% of GPs/CNS with interest in Menopause (Figure 5c).

With regard to endometriosis severity, 64.1% of responders prescribe combined HRT to all regardless of severity or completeness of excision, 15.5% prescribe it only if completeness of excision is uncertain and 6.3% only in cases of severe endometriosis (stage III/IV). The remaining 14.1% either do not prescribe combined HRT (7.1%) or individualize their HRT prescription (7.0%) (Figure 5c). On the other hand, 8.5% prescribe EO-HRT to all regardless of severity or completeness of excision, 15.7% only if excision is complete, 22.2% only in mild endometriosis or isolated adenomyosis and 7.2% individualize their HRT prescription. The remaining 46.5% would not prescribe OE-HRT regardless of severity or completeness of excision of endometriosis (Figure 5d).

#### 3.2.4. Rational and Evidence for Prescribing Combined HRT

Most responders offer combined HRT to prevent recurrence of endometriosis (79%), development of de novo endometriosis (39%) or malignant transformation of residual endometriosis (42%). Sixty-three percent of responders prescribe combined HRT based on anecdotal evidence and personal experience. On the other hand, 54% of responders indicated the lack of sufficient evidence and 66% lack of clear guidance for the use of HRT in this group of women.

#### 3.2.5. Timing of HRT after Surgery

Overall, 63.4% of the responders start HRT immediately after surgery, 2.6% after 1–4 weeks, 17.0% after 6 weeks, 11.1% after 3–6 months and 5.2% after varied durations. There were variations in the timing of HRT after surgery according to the grades of responders. The percentages of responders starting immediately were 50% for endometriosis specialists, 46% for General Gynecologists, 70% for Gynecologists with interest in menopause and 65% for GPs. While the percentages of responders starting 6 weeks after surgery were 21% for endometriosis specialists, 25% of General Gynecologists, 15% for Gynecologists with interest in menopause and 9% for GPs. The proportions of responders starting HRT 3–6 months after surgery were 7.9% for endometriosis specialists, 20.8% of Gynecologists, 15.0% for gynecologists with interest in menopause and 9.3% for GPs.

#### 3.2.6. Duration and Type of the Progestogen Component of HRT

A total of 60.9% of responders would give the progestogen component of HRT indefinitely, 19.3% for 1–2 years, 8.5% for 3–6 months, 5.3% for varied durations depending on severity of endometriosis, completeness of excision and patient’s age. The remaining 6.0% indicated that they do not prescribe combined HRT.

The type of the progestogen in HRT varied between responders with a majority prescribing micronized progesterone (48.8%), followed by Norethisterone (18.8%), Medroxyprogesterone (15.9%) and other forms, e.g., COC, combined patches and Tibolone (16.5%).

#### 3.2.7. The Oestrogen Component

Most responders prescribe transdermal oestrogen (73.0%), either as patches (36.8%), gel (27.6%) or unspecified (8.6%). A total of 24.3% stated that they have no preference and let women decide. Only 2.7% of the responders indicated that they prescribe oral preparations (<1%), implant (<1%) or Tibolone (1.3%).

#### 3.2.8. Incidence of Endometriosis Recurrence after Pelvic Clearance

Overall, 56.6% of responders stated that they see recurrence of endometriosis in <5% of women after pelvic clearance, 18.4% in 5–10%, 5.3% in 11–20% and 3.3% in >20%. The remaining 16.5% stated that they are not sure.

#### 3.2.9. HRT Alternatives and Lifestyle Measures

Only 24.3% of the responders provide all women with information for alternative HRT treatments such as herbal therapy and acupuncture, while 64.5% provide the information only when HRT is contraindicated. The remaining 11.2% either give the information occasionally (5.3%) or not at all (5.9%).

Overall, 30.8% of responders provide advice with full information on holistic and lifestyle measures, while 65.0% only offer general advice. The remaining 4.2% would only discuss this approach if HRT is contraindicated.

## 4. Discussion

In this study, we have firstly reported a case of recurrence of severe endometriosis symptoms with loss of right renal function five years after STH-BSO for severe endometriosis whilst on a relatively high dose EO- HRT. It is possible that the right renal failure was due to peri-ureteric endometriosis, although surgical injury remains another possible mechanism. Notably, the pelvic clearance did not involve a complete excision of endometriosis as it was a subtotal hysterectomy leaving behind a cervical stump with surrounding deep endometriosis. Interestingly, she remained well and pain free for around three years before the pelvic pain started to recur and progressively worsened to extreme levels. Although, addition of progestogen to her HRT gave some relieve of the pain, her original physician stopped it indicating that it was not needed as she has had a hysterectomy. Complete excision of the cervix with all surrounding endometriosis resulted in complete relieve of pain.

Secondly, and to the best of our knowledge, we report the first national survey on HRT practice in women with endometriosis after surgical menopause. The survey involved 216 physicians including endometriosis and/or menopause experts with vast experience in treating women with menopause after pelvic clearance for endometriosis. The results show a wide variation in practice for prescribing HRT in this group of women. Despite the recent guidelines by BMS and BSGE advising against OE-HRT, only two thirds of our expert responders would give combined HRT in addition to 11% who would give Tibolone. The timing of HRT after pelvic clearance and the duration of the progestogen component varied amongst responders.

Notably, there was a difference in practice between different grades of responders. For instance, while ~70–81% of endometriosis specialists, GPs/CNSs with interest in Menopause and general Gynaecology consultants prescribe combined HRT to all, only 45% of Gynaecology consultants with interest in menopause would prescribe it. Similarly, the least group to prescribe EO-HRT is GPs/CNSs (4.7%), while the highest was Gynaecology consultants with interest in Menopause (25%).

### 4.1. Results in the Light of Previous Research

Our case report is consistent with several previous case reports. In a systematic review, Gemmell et al. found that 13 out of the 14 reported cases of endometriosis recurrence after surgical menopause were receiving EO-HRT, while only one was on combined HRT [8]. Notably, seven of these 14 reported cases had extensive or severe endometriosis. Furthermore, the time lag (4 years) in our case between hysterectomy and the presentation with endometriosis recurrence was similar to the previously reported time (median 7.1 years) in the 14 case reports [8].

Our case report is also in agreement with a previous retrospective cohort study including 90 women showing recurrence of endometriosis only in women receiving EO-HRT (4/50, 8%) compared to no recurrence in women taking combined HRT (0/40) [11]. On the other hand, a clinical trial by Matorras et al. reported that endometriosis recurrence was seen after six months from surgical menopause in 3.5% (4/115) in women receiving combined HRT versus 0% (0/57) of women not on HRT [12]. Notably, two of the 4 cases of recurrence had subtotal hysterectomy or BSO alone (22%, 2/9). In contrast, Acien et al. reported no recurrence of endometriosis during 4.3 years of follow up in 11 women receiving combined HRT and eight women receiving no HRT [13].

### 4.2. Interpretation of Results

It is clear from our case report that inadequate endometriosis surgery with incomplete disease excision followed by EO-HRT pose a real risk of progression of residual endometriosis into potentially severe disease. Another important risk to be considered is malignant transformation of residual disease, especially when exposed to high dose EO-HRT for a long time. It is important to reiterate here that hysterectomy alone will not cure endometriosis, but the complete eradication of disease. Given the complexity of our case, the initial surgery should have been better performed by an endometriosis specialist to ensure adequate surgery. Subtotal hysterectomy should be avoided in these cases.

HRT was necessary in our case after her initial surgery to protect bone health in the long-term and to relieve severe menopausal symptoms. However, EO-HRT should have been completely avoided to prevent aggravation and progression of the residual disease. Instead, continuous combined HRT should have been given as recommended by current guidelines from BMS and ESHRE [9,10]. Furthermore, it was necessary to investigate and monitor for possibility of malignancy while on high dose EO-HRT, especially when the MRI revealed the complex peri-cervical endometriomas.

EO-HRT may only be considered in premenopausal women after surgical menopause in cases with mild endometriosis or solitary adenomyosis. Even in these cases, it may be sensible to give C-HRT to minimize the risk of pain recurrence.

Progression of disease on EO-HRT in our case can be explained by the well-established oestrogen dependence of endometriotic lesions. Molecular studies have revealed the overexpression of oestrogen receptor beta (ERβ), which is >100 times higher in endometriotic lesions compared to normal endometrial tissue. Furthermore, the highly expressed oestrogen in endometriotic tissue is present in its most biologically active form [14,15].

With regard to the survey, it was interesting to see that around 21% of the responders would not prescribe C-HRT or Tibolone (Figure 3a). It is possible that some physicians do not prescribe C-HRT due to lack of good evidence to support its use. Others may argue that C-HRT is only required in women with severe disease, especially if it has not been completely excised. It is also possible that some gynecologists and GPs may not be aware of the recent updates of the guidelines by BMS and ESHRE. Furthermore, some physicians would avoid progestogens due to associated increase in long-term risk of breast cancer. However, current evidence suggests that the use of progesterone with oestrogen is not associated with any long-term increase in breast cancer risk, while the use of progestins carries 16–69% increased risk (depending on the type of progestin) compared with HRT never-use [16,17].

### 4.3. Limitations and Strengths

Our aim was to target all members BSGE and BMS in addition to a representative sample of other Gynaecology consultants who manage enough numbers of women with surgical menopause due to endometriosis. Although the number of responders in each of these groups ware not as large as we hoped, they were a very good representative sample. Furthermore, all the responders were very experienced in the condition addressed in the survey (Figure 3b). Additionally, our completion rate of 70% is considered excellent in this type of survey. Another strength of this survey is that it was reviewed and approved by representatives of committees of BSGE and BMS, who have kindly distributed it to their members.

Like most surveys, it is possible that our survey relied on self-reporting of practice with no independent verification of actual responder’s management of the given scenarios. Although responders were asked what they currently do and not what they think they should do, it is possible that being presented with a list of ‘ideal’ options may have resulted in respondent bias.

### 4.4. Implications for Clinical Practice

Given the real risk of recurrence of potentially severe endometriosis whilst on EO-HRT, we strongly recommend continuous combined HRT following pelvic clearance for endometriosis, especially in severe cases. Tibolone may also be considered as an alternative to combined HRT, although there is currently no evidence to support its efficacy and safety in this group of women.

With regard to timing of HRT after pelvic clearance with complete excision of endometriosis, limited available short-term data suggest that immediate start of HRT seems to be safe with no increased risk of endometriosis recurrence [8]. Concerning the duration of the progestogen component of combined HRT, it would be advisable to continue this indefinitely in view of the previously reported long time lag (up to 13 years) between surgery and onset of recurrence of endometriosis [8]. Although there is currently limited evidence for the duration of the progestogen component of HRT, both BMS and ESHRE guidelines recommend C-HRT until natural age of menopause [9.10]. As mentioned above, although long-term use of progestins is associated with a small increase in the risk of breast cancer, the use of progesterone seems to be free of that risk [16,17].

### 4.5. Implications for Research

Most survey responders (63%) indicated that their decision when prescribing HRT in this group of women is based on anecdotal evidence and personal experience implying the lack of sufficient evidence in that area. The recent guidelines of both BMS and ESHRE acknowledge the limited evidence with regard to the optimal HRT regimen in women with endometriosis. There is therefore a need for an appropriately designed and sufficiently powered studies to address this gap in evidence. Randomised trials may be challenging in this area, but prospective cohort studies may be more feasible and adequate to provide answers to all research questions. These should investigate the appropriate type and dosages of HRT (combined HRT, EO-HRT or Tibolone) in relation to severity of disease, completeness of excision and age of patient. Other important questions include the optimal timing of HRT after pelvic clearance and the duration of the progestogen component in combined HRT. It will be essential that studies have long follow up periods to answer all the important questions.

## 5. Conclusions

The reported case clearly highlights the real risk of recurrence of potentially severe endometriosis when estrogen only HRT is used. The survey highlights the wide variation in practice when using HRT in women with endometriosis. It also shows that only two-thirds of responders prescribe combined HRT despite the recent guidelines advising against EO-HRT. In addition to raising awareness, the survey highlights the lack of evidence and the need for more research.

## Figures and Tables

**Figure 1 jcm-12-00336-f001:**
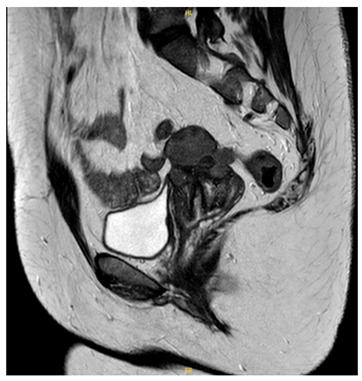
Pelvic MRI image with a lateral view showing multiple endometriomas around the cervical stump with extensive adhesions to the sigmoid colon.

**Figure 2 jcm-12-00336-f002:**
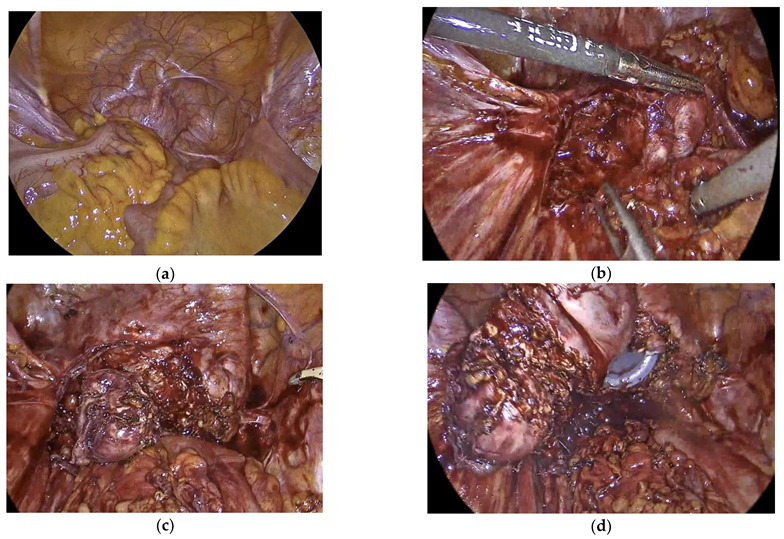
Laparoscopic images of extensive adhesiolysis and excision of peri-cervical endometriomas and cervical stump (**a**) Pelvic survey showing complete obliteration of pelvis with sigmoid adhesions; (**b**) Peri-cervical endometriomas revealed after complete mobilization of bowel; (**c**) Cervical stump completely dissected from surrounding adhesions and bowel; (**d**) Colpotomy with complete excision of cervix with the attached endometriomas.

**Figure 3 jcm-12-00336-f003:**
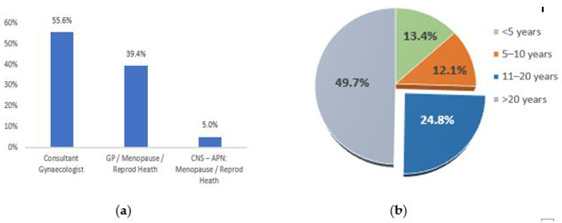
Grades and experiences of responders to the Survey (**a**) Percentages of different physicians/practitioners who responded to the survey; (**b**) Percentages of responders with different years of experience in managing women with surgical menopause after pelvic clearance for endometriosis. Abbreviations: GP, General Practitioner; CNS, Clinical Nurse Specialist; APN, Advanced Nurse Practitioner.

**Figure 4 jcm-12-00336-f004:**
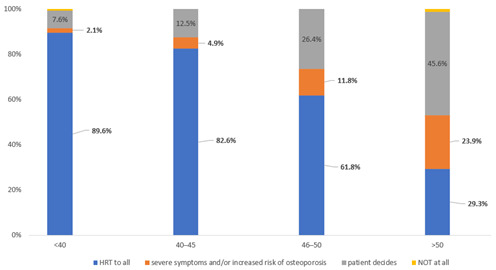
HRT and women’s age: the percentages of responders prescribing HRT to different age groups after surgical menopause in women with endometriosis.

**Figure 5 jcm-12-00336-f005:**
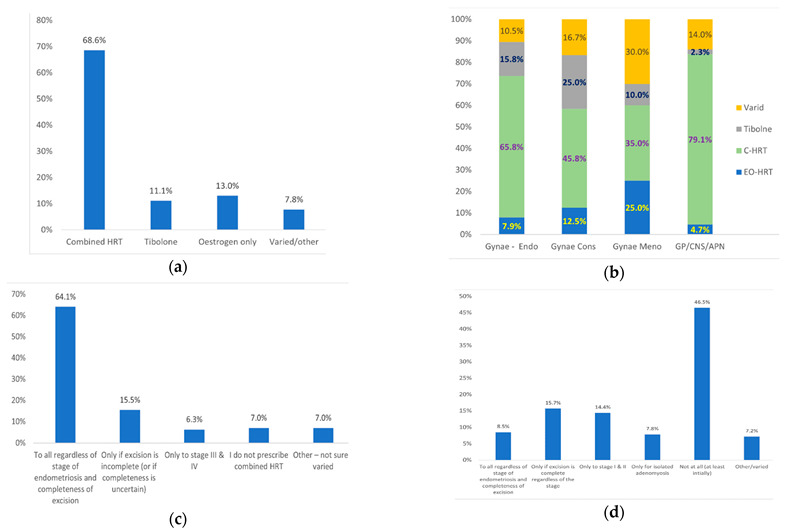
Type of HRT prescribed (**a**) Overall percentages of responders prescribing different types of HRT; (**b**) Percentages of responders of different grades prescribing different types of HRT; (**c**) Percentages of responders prscribing combined HRT according to endometriosis severity or completeness of excision; (**d**) Percentages of responders prescribing oestrogen only HRT according to endometriosis severity or completeness of excision. Abbreviations: Gynae, Gynaecology; Endo, Endometriosis Specialist; Cons, Consultant; Meno, Menopause specialist; GP, General Practitioners; CNS, Clinical Nurse Specialist; APN, Advanced Nurse Practitioner.

## Data Availability

Not applicable.

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
