# Peer review of "HRT in Women Undergoing Pelvic Clearance for Endometriosis—A Case Report and a National Survey"

_jcm, 2023, doi:10.3390/jcm12010336_

Round 1
Reviewer 1 Report
Τhis is a very ineresting study that demonstrates gaps in knowledge as well as in daily practice in the use ofHRT in women who have undergone surgery for endometriosis.
comment 1
I suggest that the authors add a paragraph to the introduction where they mention the wide range of symptoms as well as the areas of appearance of endometriosis. there are extremely rare cases of endometriosisthat should be referred. the following rare entities are mentioned indicatively.
Ait Benkaddour Y, El Farji A, Soummani A. Endometriosis of the vesico-vaginal septum: a rare and unusual localization (case report). BMC Womens Health. 2020 Aug 14;20(1):179. doi: 10.1186/s12905-020-01047-w. PMID: 32795369; PMCID: PMC7427748.
Papageorgiou D, Prantalos P, Tzavoulis D, Sgouros SN, Ivros N, Papakonstantinou K. Massive hemorrhagic ascites associated with extensive severe peritoneal endometriosis: A rare case report. Int J Gynaecol Obstet. 2022 Jul;158(1):216-217. doi: 10.1002/ijgo.14148. Epub 2022 Feb 28. PMID: 35182071.
Papageorgiou D, Ivros N, Kalles V, Papapanagiotou IK, Papakonstantinou K. Hemorrhagic hydrocele in the canal of Nuck: A rare case of endometriosis. Eur J Obstet Gynecol Reprod Biol. 2021 Sep;264:382-383. doi: 10.1016/j.ejogrb.2021.07.011. Epub 2021 Jul 17. PMID: 34304934.
Xavier J, Buckland B, Stewart P. Rare case of spontaneous sub-umbilical endometriosis within an umbilical hernia in a 39-year-old female. ANZ J Surg. 2020 May;90(5):895-896. doi: 10.1111/ans.15352. Epub 2019 Jul 23. PMID: 31338940.
Author Response
Please attached file

Reviewer 2 Report
The authors present an interesting and debatable case of a woman who had a subtotal hysterectomy and bilateral salpingo-oophorectomy with incomplete debulking of endometriosis and disease progression after having been on oestrogen-only HRT for five years. The authors highlight the absence of evidence behind the use or the different options of HRT to treat menopausal and endometriosis-related symptoms after surgically-induced menopause, and present the results of a national survey conducted in the United Kingdom among a group of doctors with different levels of expertise.
Although this is a topic of interest, there are some aspects of the manuscript that need significant improvement, particularly the discussion. Please review the attached file with all of my comments.
